

# Trunk variability and local dynamic stability during gait after generalized fatigue induced by incremental exercise test in young women in different phases of the menstrual cycle

Ludmila Dos Anjos[1], Fábio Rodrigues[1], Sofia Scataglini[2], Rafael Reimann Baptista[3], Paula Lobo da Costa[4] and Marcus Fraga Vieira[1]

[1] Bioengineering and Biomechanics Laboratory, Federal University of Goiás, Goiânia, Brazil
[2] Department of Product Development, Faculty of Design Science, University of Antwerp, Antwerp, Belgium
[3] School of Health and Life Sciences, Pontifícia Universidade Católica do Rio Grande do Sul, Porto Alegre, Brazil
[4] Department of Physical Education, Federal University of São Carlos, São Carlos, Brazil

## ABSTRACT

**Purpose**. The purpose of this study was to identify how generalized fatigue along with hormonal changes throughout the menstrual cycle affects trunk variability and local dynamic stability during gait.

**Methods**. General fatigue was induced by an incremental test on a treadmill, and the menstrual cycle was divided into three phases: follicular, ovulatory, and luteal. Twenty-six healthy, young volunteers (aged 18 to 28 years) who did not use oral contraceptives or other hormonal drugs with a regular menstrual cycle participated in the study. They walked on the treadmill for 4 min at the preferred speed, before the incremental test, followed by four sets of 4 min alternating between walking, also at preferred speed, and resting. From trunk kinematic data, the following were extracted: the mean of the standard deviation along strides, as a measure of variability, and the maximum Lyapunov exponent, as a measure of local dynamic stability (LDS).

**Results**. After the incremental test, variability increased, and LDS decreased. However, they showed a tendency to return to the initial value faster in women compared to previous results for men. In the follicular phase, which has less hormonal release, the volunteers had an almost complete recovery in LDS soon after the first rest interval, suggesting that female hormones can interfere with fatigue recovery. Nevertheless, concerning the LDS, it was significantly lower in the luteal phase than in the follicular phase.

**Conclusion**. Women that are not taking oral contraceptives should be aware that they are susceptible to increased gait instabilities in the pre-menstrual phase after strenuous activities.

Corresponding author
Rafael Reimann
Baptista, rafael.baptista@pucrs.br

## INTRODUCTION

Women produce and release gametes in monthly cycles (average 28 days, normal range of 24–35 days) divided into three phases (*McNulty et al., 2020*): follicular, ovulatory, and luteal phases. During the follicular phase of the cycle, estrogen is the dominant steroid hormone, the ovulatory phase is triggered by the peak of the luteal and follicular stimulant hormones, and in the luteal phase progesterone is dominant, although estrogen is still present (*McNulty et al., 2020*). During the menstrual cycle, the responses of the central nervous system to female hormones may vary as they depend on the hormone concentration levels (*Darlington et al., 2001*).

Some experimental findings support this hypothesis. *Cabelka et al. (2019)* found that the injection of progesterone and estrogen into rats possibly increased fatigue resistance in these animals. *Schneider et al. (2004)* injected progesterone and estradiol in mice to measure the time-fatigue effects of eccentrically contracted plantar flexor muscles and the isometric torque of such muscles immediately after the protocol of eccentric contraction. Plantar-flexor muscle fatigue during eccentric contraction occurs later in the progesterone group than in the estradiol group and the combined estradiol and progesterone group. The progesterone group exhibited the highest isometric torque immediately after the eccentric contraction protocol. These findings suggest that progesterone reduces muscle fatigue in response to eccentric contractions, and this effect is decreased when estradiol is also present (*Schneider et al., 2004*). Then, even with different protocols and recovery times, the intake of progesterone seems to be beneficial in fatigue recovery. The hormonal fluctuation during the menstrual cycle can possibly affect the response of women to fatigue.

Exercises involving the whole body, such as running, cycling, or incremental exercise test, induce general fatigue (*Paillard, 2012*), which deteriorates sensory information and decreases muscle force (*Lepers et al., 1997*; *Nardone et al., 1998*; *Paillard, 2012*). In addition, general muscular exercise induces dehydration, provoking alterations in vestibular function (*Derave et al., 1998*), important for gait/running balance and stability. Dehydration alters blood viscosity and compromises blood flow to the inner ear structures responsible for balance control (*Smith, Kovacs & Pearce, 2013*), can also disrupt electrolyte balance in the body, particularly in situations of prolonged or intense exercise (*Kleinsasser et al., 2005*), and alters brain activity and connectivity patterns, particularly in regions associated with vestibular processing (*Saker, Farrell & Adib, 2019*), which together can disrupt the normal functioning of the vestibular system, leading to a decline in balance control and spatial orientation.

Prolonged and repetitive vertical and lateral movements during exercises such as running or walking provoke exacerbated stimulation of visual input and alter otolithic sensitivity, decreasing the quality of vestibular information to the motor control (*Derave et al., 2002*). Furthermore, prolonged eccentric muscle contractions during running produce more muscle damage and soreness that alter proprioception and deteriorate the sense of limb position and force (*Nardone et al., 1997*; *Paschalis et al., 2007*; *Vissing et al., 2008*).

Mental fatigue, arising from prolonged cognitive activity or demanding mental tasks, has been found to have a detrimental impact on physical performance. Research has shown

that mental fatigue can lead to reduced endurance, impaired motor coordination, and decreased muscular strength. Studies conducted by *Marcora (2009)* have demonstrated that individuals experiencing mental fatigue exhibit decreased time to exhaustion during endurance exercises, such as cycling or running. Additionally, studies investigating motor coordination tasks, such as balance control and precision movements, have reported compromised performance following mental fatigue (*Brisswalter, Collardeau & René, 2002*; *Boksem, Meijman & Lorist, 2006*).

However, there are a lack of studies assessing the effects of general or local fatigue on trunk stability during gait in women. Previous studies have focused on a male population (*Barbieri et al., 2013*; *Vieira et al., 2016*), or did not mention whether women were included in the assessed groups (*Hamacher et al., 2016*). Furthermore, in previous studies examining unilateral induced fatigue of hip abductor (*Arvin et al., 2015*), or on unilateral leg muscle fatigue (*Toebes et al., 2014*), changes in gait local dynamic stability (LDS), quantified by Lyapunov exponent, were not reported. On the other hand, studies assessing general fatigue effects on gait stability have produced contrasting results concerning the Lyapunov exponent if the used test was an incremental maximal exercise test (IMET) on a treadmill (*Vieira et al., 2016*) or on a cycle ergometer (*Hamacher et al., 2016*).

Therefore, we hypothesized that (1) gait LDS will decrease, and gait variability will increase just after IMET; (2) the recovery period will be equal or shorter than 20 min for the participants, given that 20 min had not been enough for a complete recovery in previous studies with men (*Vieira et al., 2016*); (3) gait LDS, gait variability, and recovery period can be affected by the phases of the menstrual cycle, with the best results in the follicular phase, where estrogen and progesterone concentrations are low, the worse results in the luteal phase, in which the premenstrual symptoms starts and gait is presumed to be influenced by the increase of progesterone and estrogen, and with the results in the ovulatory phase between them. Considering the aspects aforementioned and the lack of women included in performance fatigability studies (*Hunter, 2016b*), the present study aims to analyze gait variability and LDS before and immediately after IMET on a treadmill, and during the recovery period after IMET, in young and healthy women during the different phases of the menstrual cycle.

## METHODS

### Study design

This is an experimental and descriptive study in which the participants walked before and immediately after IMET on a treadmill, and during the recovery period after IMET, in three sessions, in which kinematic data were collected, corresponding to the three phases of menstrual cycle.

### Participants

Twenty-six young women aged 18 to 28 years (Table 1) took part in the study. They were healthy, nulliparous, non-smokers, not sedentary according to the classification of International Physical Activity Questionnaire (IPAQ) short version (*Craig et al., 2003*), did not use contraceptive pills, and had a regular menstrual cycle for at least six months

**Table 1  Characteristics of the participants.**

| Characteristic | N = 26 |
|---|---|
| Height (cm) | 162.66 ± 8.22 |
| Age (years) | 21.86 ± 2.67 |
| Body mass (kg) | 54.14 ± 15.35 |
| BMI (kg/m$^2$) | 20.15 ± 3.62 |
| Regular cycle (days) | 28.0 ± 1.53 |
| Duration of mense (days) | 4.86 ± 1.21 |

Notes.
Values are expressed as mean ± standard deviation.
N, number of participants; BMI, Body Mass Index.

(concerning duration, in number of days, and presence of ovulation). The following exclusion criteria were established: injury in the lower limbs in the last six months; the presence of blisters or cutaneous lesions on the feet; history of balance dysfunction; obesity; does not have a normal eumenorrheic menstrual cycle; menarche for less than two years; smokers; cardiac and/or respiratory pathologies.

All protocols were approved by the Federal University of Goias Ethics Committee for Human Research (protocol number 3.913.182), and the participants signed an informed consent form.

## Protocol

The participants were followed up for two months before data collection, to assure a regular menstrual cycle, although they had reported menstrual cycle regularity in the last six months. During this period, the duration of the menstrual cycle, date of menses, length of menses, and basal body temperature immediately after waking up were recorded using a mobile application (Menstrual Calendar Paula, Famivita, Sao Paulo, Brazil) for menstrual cycle phase estimation.

The participants walked before and immediately after IMET on a treadmill, and during the recovery period after IMET, in three sessions corresponding to the three phases of menstrual cycle: early follicular (days 1–5, based on menses onset), ovulatory (days 13–15, based on basal body temperature), and mid-luteal (days 20–23) (*McNulty et al., 2020*), with a tolerance of two days. The fatigue protocol for kinematic data collection was based on a previous study (*Vieira et al., 2016*) with an extended recovery period as shown in Fig. 1.

Reflective markers were attached to the heels, the lateral malleoli, the head of second metatarsals, for step identification (*Souza et al., 2017*), and the first thoracic spinous process (T1), for trunk kinematics. The reflective markers were used for kinematic data collection using a 3D motion capture system comprising ten infrared cameras operating at 100 Hz (Vicon, Oxford Metrics, Oxford, UK).

All participants were used to walking on a level treadmill. Despite this, they performed a familiarization walking for at least two minutes. Next, the preferred walking speed on the treadmill was estimated according to a previously reported protocol (*Dingwell & Marin, 2006*). After 1-min rest, the participant walked on the treadmill at preferred walking speed for 4 min (Pre-Test, PreT). Immediately after the Pre-Test, the participants were

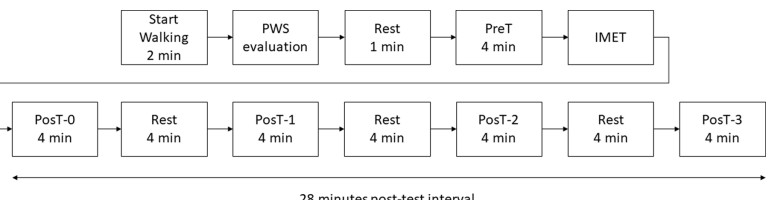

**Figure 1** Flowchart of the protocol used in the study.

submitted to an adapted IMET on the treadmill: the speed was adjusted to 6 km/h, followed by progressive increments of the treadmill speed at a rate of 1 km/h every minute until participant exhaustion (the participant asked to stop).

For testing the recovery of the outcomes, *i.e.,* the return to basal values at PreT, immediately after the IMET, the participant executed a 4-min walking at preferred walking speed (PosT-0), followed by three sequences of 4-min of rest (seated) and 4-min of walking at preferred walking speed (PosT-1, PosT-2, and PosT-3), until the end of the protocol at 28 min (Fig. 1).

The participants wore a wearable heart rate chest-strapped measurement system (Polar H10) during each experimental session. The subjective exertion level was assessed through the Borg 0–20 scale of perceived exertion (*Borg, 1990*). The exercise was considered a suprathreshold when the heart rate exceeded 60% of the maximum heart rate for each participant, estimated as 220–age (years)–0.6 (*Wasserman, 1984*).

After the last session, the participants reported the onset of menses in the following menstrual cycle, although they presented a regular menstrual cycle in the last six months. This procedure was adopted to confirm the regularity of the menstrual cycle.

## Data analysis

The kinematic data were filtered with a fourth-order, zero-lag, low-pass Butterworth filter, with a cutoff frequency of 10 Hz, except for the LDS assessment, for which the data was filtered with a cutoff frequency of 25 Hz. For each trial of each session, the initial and final 15 s of each test were discarded (*Hak et al., 2013*). Then, all steps were detected as the zero crossing of the speed of the heel markers in the anterior-posterior direction (*Zeni, Richards & Higginson, 2008*; *Souza et al., 2017*). Next, the 150 intermediate strides were selected, discarding the initial and final strides.

Because balance and posture control are essential mechanisms in locomotion (*Dimitrijevic & Larsson, 1981*), and maintaining the stability of the upper body is critical for human locomotion (*Prince et al., 1994*), the analyzed variables were extracted from trunk kinematics. Gait variability was evaluated by the standard deviation of the trunk acceleration along the strides, and trunk local dynamic stability was assessed extracting the maximum Lyapunov exponent from trunk velocity. A customized Matlab (R2018a, MathWorks, Natick, MA, USA) code was used for data analysis.

*Gait variability*

To calculate trunk variability (VAR), each stride was normalized to 101 points (0–100%). At each of the 101 points, the standard deviation of the trunk acceleration in the medial-lateral (ML), anterior-posterior (AP), and vertical (V) directions over the 150 selected strides were calculated. Then, the average value of these 101 standard deviations was calculated (*Dingwell et al., 2001*).

*Local dynamic stability*

Local dynamic stability was quantified by computing the maximum Lyapunov exponent ($\lambda$) from medial-lateral, anterior-posterior, and vertical trunk velocity, using Rosenstein's algorithm (*Rosenstein, Collins & de Luca, 1993*). The 150 strides time series were time-normalized to 15,000 samples, preserving the fluctuation in stride time between strides (*Stenum, Bruijn & Jensen, 2014*; *Raffalt et al., 2019*). Next, a high-dimensional attractor was constructed using the normalized trunk velocity and its delayed copies. An individualized delay ranging from 8–12, 9–14, and 10–13 samples was adopted for medial-lateral, anterior-posterior, and vertical directions, respectively, based on the mean value of the minimum of the mutual information function (*Fraser, 1986*) across all participants, and an average dimension of 8, 6 and 7 for medial-lateral, anterior-posterior, and vertical directions, respectively, was adopted across all participants based in the false nearest neighbor analysis (*Kennel, Brown & Abarbanel, 1992*). For each point in the entire state-space, the nearest neighbor was found, and the Euclidean distance between these points was tracked for a time interval corresponding to approximately ten strides, resulting in time-distance curves to each time point in the state-space (for a more detailed algorithm's description see Appendix A in *Terrier & Dériaz, 2013*). The $\lambda$ was calculated as the slope of a linear fit to the first 50 samples (average time needed for one step) of the mean divergence curve, obtained as the mean of the natural log of the time-distance curves and expressed as ln(div)/stride-time.

## Statistical analysis

As the analyzed variables presented a normal distribution (Shapiro–Wilk test, $p > 0.05$), repeated measures analysis of variance (ANOVA) with two factors was conducted to assess the main effect of menstrual cycle phases, the main effect of trials, and the interaction effect. When significant main and/or interaction effects were found in the ANOVA, post-hoc tests with Bonferroni correction were conducted to assess differences between the trials, and between menstrual cycle phases. The statistical tests were performed using JASP software, version 0.15.0 (https://jasp-stats.org/), with a significance level set at $\alpha < 0.05$.

## RESULTS

The resting heart rate and the maximum heart rate obtained after the incremental test are shown in Table 2, as well as the Borg scale and the maximum running speed. The heart rate (Table 2) showed significant difference only between the pre-test (resting) and post-test (maximum) trials ($p < 0.001$), showing no difference between menstrual cycle phases ($p = 0.138$), and no interaction phase *vs* trial effect ($p = 0.912$).

**Table 2** Heart rate before and after Incremental Test, effort perception, and walking speed ($N = 26$ volunteers).

| | Menstrual cycle phases | | | $p$ |
|---|---|---|---|---|
| | **FP** | **OP** | **LP** | |
| Resting heart rate (bpm) | $83.93 \pm 7.44^a$ | $84.00 \pm 6.11^b$ | $85.29 \pm 6.13^c$ | 0.247 |
| Maximum heart rate (bpm) | $184.14 \pm 5.87^a$ | $184.79 \pm 15.89^b$ | $186.36 \pm 5.26^c$ | 0.325 |
| Post-test effort perception (Borg) | $17.86 \pm 0.79^d$ | $17.95 \pm 0.83^e$ | $18.32 \pm 1.20^{d,e}$ | 0.038 |
| Maximum running speed (km/h) | $11.32 \pm 1.53$ | $11.4 \pm 1.28$ | $11.26 \pm 1.48$ | 0.418 |
| PWS (km/h) –overground | $4.55 \pm 0.51$ | $4.55 \pm 0.51$ | $4.55 \pm 0.51$ | # |
| PWS (km/h) –treadmill | $4.42 \pm 0.50$ | $4.42 \pm 0.50$ | $4.42 \pm 0.50$ | # |

**Notes.**
Values are expressed as mean ± standard deviation.
p, repeated measures ANOVA.
For all phases of the menstrual cycle, the heart rate is considerably greater after the incremental test (pairwise comparisons resting heart rate vs maximum heart rate: [a] (FP) - $p < 0.001$, [b] (OP) –$p < 0.001$, [c] (LP) - $p < 0.001$), but not between the menstrual cycle phases. Effort perception is greater at LP (pairwise comparisons: [d] (LP vs FP) - $p = 0.039$, [e] (LP vs OP) - $p = 0.041$). (#) PWS was the same for the different phases of the menstrual cycle.
PWS, preferred walking speed; N, number of volunteers; FP, follicular phase; OP, ovulatory phase; LP, luteal phase.

## Gait variability

In medial-lateral direction (Fig. 2A), VAR ML presented a significant main effect of trial. No main effect of menstrual cycle phases or interaction effect was observed. VAR ML was significantly higher in PostT-0 than in the remaining trials for all menstrual cycle phases ($p < 0.001$). In the follicular phase, VAR ML in PreT was significantly lower than in PosT-2 ($p = 0.025$) and PosT-3 ($p = 0.007$) trials, indicating no recovery. In the ovulatory phase, VAR ML in PreT was significantly lower than in PosT-1 ($p = 0.026$), PosT-2 ($p = 0.017$), and PosT-3 ($p = 0.003$) trials, also indicating no recovery.

In the anterior-posterior direction (Fig. 2B), VAR AP presented only a significant main effect of trial, with no main effect of menstrual cycle phases or interaction effect. VAR AP was significantly higher in PosT-0 than in the remaining trials for all menstrual cycle phases ($p < 0.001$). No other differences were observed.

In the vertical direction (Fig. 2C), VAR V presented only a significant main effect of trial, with no main effect of menstrual cycle phases or interaction effect. VAR V was significantly higher in PosT-0 than in the remaining trials, but only for follicular phase and luteal phase phases ($p < 0.001$).

## Local dynamic stability

In the medial-lateral direction (Fig. 3A), λ presented significant main effects of trials ($p < 0.001$) and of menstrual cycle phases ($p = 0.003$), with no interaction effect. PosT-0 trial was significantly higher than PreT, PosT-1, PosT-2, and PosT-3 trials for all menstrual cycle phases, with no difference between PreT and PosT-1, PosT-2 and PosT-3. Luteal phase presented significant higher values for λ than follicular phase in PreT ($p = 0.002$), PosT-1 ($p < 0.001$), and PosT-2 ($p = 0.004$) trials.

In the anterior-posterior direction (Fig. 3B), λ presented significant main effects of trials ($p < 0.001$) and of menstrual cycle phases ($p = 0.006$), with no interaction effect. PosT-0 trial was significantly higher than PreT, PosT-1, PosT-2, and PosT-3 trials for all menstrual

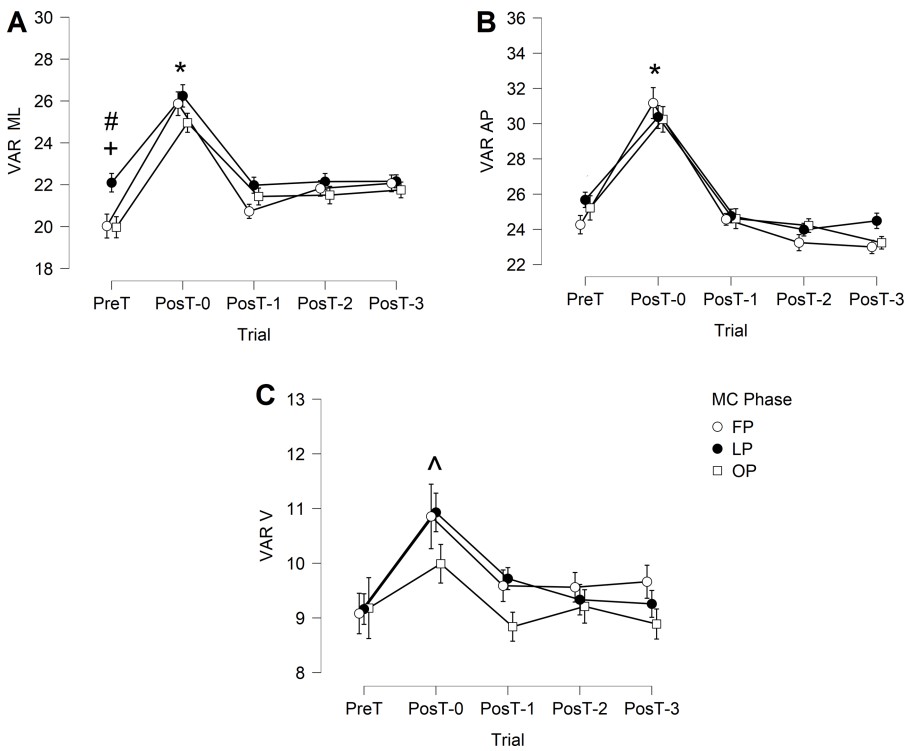

**Figure 2  Trunk variability during the gait cycle.** (A) VAR ML, mediolateral; (B) VAR AP, anteropos-terior; (C) VAR V, vertical. FP, follicular phase; LP, luteal phase; OP, ovulatory phase. * indicates signifi-cant higher PosT-0 values. + indicates significant lower PreT value compared to PosT-2 and PosT-3 trials in FP phase. # indicates significant lower PreT value compared to PosT-1, PosT-2 and PosT-3 trials in OV phase. ∧ indicates higher PostT-0 values but only FP and LP phases.

cycle phases. Additionally, in the follicular phase, PostT-1 was significantly higher than PosT-2 and PosT-3 ($p = 0.037$, $p = 0.008$, respectively). Luteal phase presented significant higher values for λ than follicular phase in PosT-2 ($p = 0.007$) and PosT-3 ($p = 0.014$) trials.

In the vertical direction (Fig. 3C), λ presented significant main effects of trials ($p < 0.001$) and of menstrual cycle phases ($p = 0.006$), with no interaction effect. PosT-0 trial was significantly higher than PreT, PosT-1, PosT-2, and PosT-3 trials for all menstrual cycle phases, with no difference between PreT and PosT-1, PosT-2, and PosT-3. Luteal phase presented significant higher values for λ than follicular phase in PreT ($p = 0.005$), PosT-0 ($p = 0.042$), PosT-1 ($0.007$), and PosT-2 ($p = 0.007$) trials, and higher than ovulation phase in PosT-1 ($p = 0.009$) trial.

Overall, the behavior of λ in the three directions ML, AP, and V were similar, with the luteal phase presenting lower local dynamic stability with higher values of λ.

## DISCUSSION

This study analyzed changes in trunk local dynamic stability and variability during gait due to generalized fatigue induced by incremental exercise test in women, based on

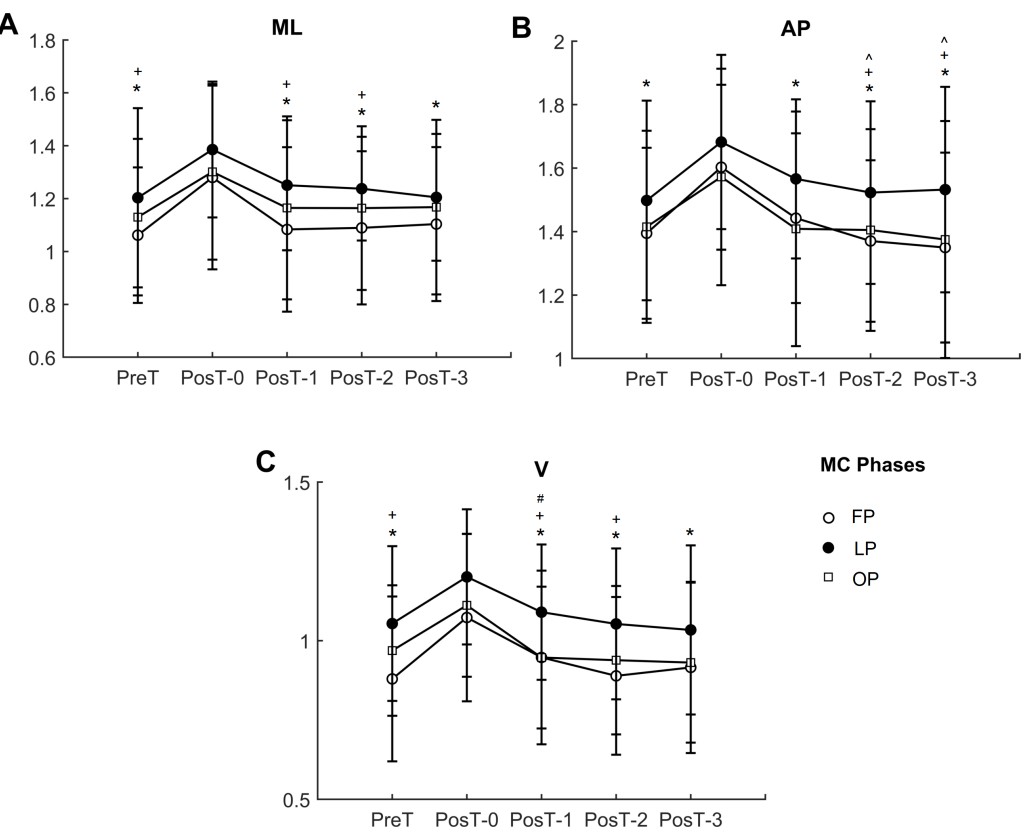

**Figure 3** **Trunk local dynamic stability in the three directions (A) ML, medial-lateral; (B) AP, anterior-posterior; (C) V, vertical directions.** FP, follicular phase; LU, luteal phase; OP, ovulation phase. * indicates significant lower value compared to PosT-0 trial; ∧ indicates significant lower values compared to PosT-1 in FP; + indicates significant higher values in LP compared to FP; # indicates significant higher values in LP compared to OP.

previous studies that have tested changes in stability due to generalized fatigue induced by incremental exercise test in men (*Vieira et al., 2016*). The main objective was to verify the effects of generalized fatigue in women and its consequent influence on trunk stability and variability during gait, and the relationship with the hormonal period of the menstrual cycle. General fatigue induced by incremental exercise test degraded gait stability and increased gait variability for all menstrual cycle phases, and there was a recovery interval shorter in women, *i.e.* the post-test outcomes (PosT) returned earlier to basal values (PreT) than that observed in previous studies with men (*Vieira et al., 2016*).

The hypotheses of the study were confirmed in part. Local dynamic stability decreased immediately after an incremental exercise test (with a significant increase of λ) for all menstrual cycle phases, a result similar to that previously reported in men (*Vieira et al., 2016*). Overall, in luteal phase the volunteers presented lower local dynamic stability (greater λ, Fig. 3) than in follicular phase (Fig. 3). Indeed, the fastest recovery was observed in follicular phase, where the estrogen and progesterone concentrations are lower. However, although the volunteers presented a slightly better performance in follicular phase, such

result was only significant when compared to luteal phase. Although previous studies had not reported any significant differences in aerobic capacity and exercise responses during any phase of the menstrual cycle (*Constantini, Dubnov & Lebrun, 2005*), some studies suggest a possible slight decrease in aerobic capacity (*Lebrun & Rumball, 2001*) or exercise response during the luteal phase (*Freemas et al., 2021*), which may explain the differences above. The recovery time for the follicular phase was less than half the time tested, with the $\lambda$ in the medial-lateral and anterior-posterior directions returning to the PreT value at the PosT-1 instant (Fig. 3), although PosT-2 and PosT-3 instants presented lower values than PosT-1 instant. Furthermore, the expectations for ovulatory phase, a phase that has high estrogen concentration and low progesterone release, were not met, since it was hypothesized that the recovery would not be as fast as in follicular phase. In ovulatory phase, $\lambda$ returned to its pre-test values at the same instants (Fig. 3) compared to follicular phase.

The increase of progesterone in luteal phase influenced LDS but not the recovery time. Overall, as mentioned, luteal phase presented the lowest LDS (greatest $\lambda$, Fig. 3), however, the expected recovery time for $\lambda$ was the same as in follicular phase (Fig. 3), although it was better in women than in our previous findings in men (*Vieira et al., 2016*). Some experimental findings may help to explain these results, although the findings obtained in different studies, where female steroid hormones were injected into mice, were somewhat conflicting and could not be directly extended to humans. For local muscle fatigue, there was a faster recovery in the presence of progesterone than in the presence of progesterone and estrogen or estrogen alone (*Schneider et al., 2004*). However, in another study, progesterone and estrogen together proved to be more effective in recovering fatigue (*Cabelka et al., 2019*).

Women, who generally have lower muscle mass and absolute muscle strength, may exhibit greater resilience than men in submaximal, isometric, and dynamic tasks due to their lower vascular compression and muscle oxygen demand (*Enoka & Stuart, 1992*; *Hunter, 2014*; *Hunter, 2016a*). They have a greater ability to use oxidative metabolism (*Russ & Kent-Braun, 2003*; *Constantini, Dubnov & Lebrun, 2005*), whereas men have a greater dependence on glycolytic pathways, tending to show greater impairment in neuromuscular activation after fatiguing exercise (*Constantini, Dubnov & Lebrun, 2005*; *Hunter, 2009*; *Halperin, Chapman & Behm, 2015*).

However, not only sex differences in muscle mass can influence these responses, gender differences in the cardiovascular system, including ventricular size, lung size, blood volume, and hematocrit, plays an important role in muscle efficiency as a larger cardiovascular system can provide more oxygen and nutrients to the muscles, allowing them to perform more efficiently (*Mitchell et al., 1994*; *Bassett & Howley, 2000*). Additionally, the number and size of mitochondria within muscle cells can also impact muscle efficiency, as mitochondria are responsible for generating energy within the cell (*Hood, 2001*).

Furthermore, genetic factors can determine an individual's muscle fiber type and distribution, which can impact their ability to perform certain types of exercise (*Gollnick et al., 1972*). Age can also impact muscle efficiency, as older individuals tend to experience declines in muscle mass and function (*Lexell, Taylor & Sjöström, 1988*). Finally, training

status can have a significant impact on muscle efficiency, as regular exercise can improve muscle strength, endurance, and overall performance (*Sale, 1988*).

Although *Vieira et al. (2016)* did not compare genders, their findings have shown that, for men, the post-fatigue recovery is slower, with the analyzed variables showing a tendency to return to their pre-test values at the instant PosT-2, while women in the current study recovered faster (Fig. 3), mainly in follicular phase, where the return to PreT occurred at the instant PosT-1, a trial just after the first rest interval. Thus, it was observed that in the period of less hormonal release, a period more comparable to men who have a practically stable hormonal system, women tend to recover faster after strenuous treadmill exercise.

According to the Borg scale of effort perception results of the present study (Table 2), the highest value of perceived exertion occurred in the luteal phase. Although the results reported in the literature concerning the effects of menstrual cycle phases on women exercise performance in women are controversial, some studies suggest that progesterone stimulates ventilation regardless of running intensity, which may increase the perception of exertion, since runners often link their perception of exertion to how much they are breathing (*Karp & Smith, 2012*). Breathing rate increases during the luteal phase when the progesterone concentration is higher, making women feel more tired during training in luteal phase than in follicular phase (*Constantini, Dubnov & Lebrun, 2005*; *Karp & Smith, 2012*; *Freemas et al., 2021*).

In the present study, heart rate did not present a significant difference between the phases of the menstrual cycle (Table 2), despite confirming the lower resting heart rate in follicular phase, and the higher resting heart rate in the luteal phase found in previous studies (*Bandyopadhyay & Dalui, 2012*).

The results obtained for the variability (VAR) can be interpreted in the same way as LDS. There was an increase in gait variability shortly after IMET for all phases of the menstrual cycle (Fig. 2). The high variability accompanied by a decrease in local dynamic stability after IMET may indicate less precise movement control and a higher risk of fall (*Vieira et al., 2016*), so the volunteers need to adopt a more cautious gait since a significant disturbance in gait could lead to a decrease in balance control (*Helbostad et al., 2007*). Furthermore, in the medial-lateral direction, VAR ML did not present a complete recovery in the follicular phase and ovulatory phase phases, so a longer rest interval would be necessary to have the variability returning to pre-test values. This requires special attention since theoretical and experimental studies have highlighted the importance of regulation of movement in the frontal plane: gait is considered stable in the anterior-posterior and vertical directions, but active control would be required to maintain proper control of body movement in the medial-lateral direction (*Bauby & Kuo, 2000*).

This study has some limitations. One possible limitation is regarding the treadmill familiarization time. However, since the participants were used to walking on a treadmill for training/conditioning purposes regularly, we assume that the familiarization time was enough. Another limitation is that no blood testing was done in this study to confirm the menstrual cycles. However, the participants were followed for several weeks to confirm the regularity of the menstrual phases and our findings can be used for the benefit of female athletes that are only able to control their menstrual cycles by self-evaluation.

Another possible limitation of the present study is that we did not address the influence of mental fatigue in physical performance since mental fatigue leads to a shift in the perception of effort, resulting in an increased perception of exertion during physical tasks (*Marcora, 2009*). Moreover, mental fatigue has been shown to alter neural activity within the motor cortex, affecting motor unit recruitment and muscle activation patterns (*Pageaux, 2014*). Finally, the treadmill walking design imposes some limitations, as it is well established in biomechanics that treadmill walking reduces gait variability and increases gait local dynamic stability (*Dingwell et al., 2001*; *Shi et al., 2019*; *Rosenblatt & Grabiner, 2010*; *Hollman et al., 2016*; *Yang & King, 2016*; *Toro et al., 2022*), so that such variables may have been underestimated. However, a more controlled design is necessary as the calculation of these variables requires a large number of gait cycles, as well as IMET requires to be executed on a treadmill.Future investigations should consider the differences between sedentary and active women, including the comparison between different exercise modalities like strength training and aerobic exercise.

## Practical implications

General fatigue induced by incremental exercise test degraded gait stability and increased gait variability for all menstrual cycle phases, and there was a recovery interval shorter than that observed in previous studies with men.

In luteal phase the volunteers presented lower local dynamic stability than in follicular phase. Indeed, the fastest exercise recovery was observed in follicular phase, where the estrogen and progesterone concentrations are lower.

The performance of tasks and sports that require intense force and resistance to exhaustion can be reduced in luteal phase compared to other phases of the menstrual cycle.

## Clinical implications

In the realm of clinical evaluation, it is crucial to recognize the significance of an Incremental Multistage Exercise Test (IMET) in assessing cardiopulmonary function. However, it is equally important to exercise caution and adopt appropriate measures following the test, particularly when dealing with patients who are susceptible to falls. In such cases, it is advised to incorporate a quiet rest interval of no less than 20 min, as this allows for the recovery of balance control and enhances gait dynamic stability.

During an IMET, various parameters are measured to gauge the individual's cardiovascular and pulmonary capacity. These tests serve as invaluable tools for healthcare professionals, enabling them to gain insights into the patient's overall fitness and identify any potential underlying issues. However, it is crucial to proceed with care after the test, especially when dealing with patients who may have a higher risk of falls.

Post-exercise recovery is a critical phase that warrants attention. For individuals prone to falls, it is recommended to provide them with a calm and tranquil environment to aid their recuperation process. Allocating a minimum of 20 min for this rest interval allows the body to regain balance control and restore gait dynamic stability, mitigating any potential risks or complications that may arise due to post-exercise fatigue.

This recovery period holds particular importance as it allows the body's physiological systems to return to a state of equilibrium. It grants the cardiovascular and respiratory

systems a chance to gradually stabilize, ensuring a smooth transition from exertion to rest. Furthermore, it aids in the restoration of muscular strength and coordination, further reducing the likelihood of accidents or falls that can occur due to fatigue-induced instability.

By emphasizing the necessity of a sufficient recovery period after an IMET, healthcare professionals can ensure the safety and well-being of their patients. The recommended 20-minute rest interval allows for a comprehensive recuperation process, enabling individuals to regain their balance control and enhance their gait dynamic stability. This approach promotes patient safety, prevents accidents, and facilitates a smoother transition from exercise to recovery, ultimately contributing to more effective clinical evaluations and improved patient outcomes.

## CONCLUSION

After the incremental test, variability increased and local dynamic stability decreased, showing a tendency to return to the initial value faster in women when compared to previously reported results in men. In the follicular phase, which has less hormonal release, the participants had higher local dynamic stability and an almost complete recovery soon after the first rest interval, suggesting that the female hormones can interfere. However, concerning the phases of the menstrual cycle, a significant difference was only observed between the follicular and luteal phases. This led to some practical applications that can be made from this study. For example, running and cycling, exercises that induce general fatigue, are among the most popular sport/fitness activities worldwide due to their simple requirements in terms of equipment and environment, being largely practiced by women. Therefore, women that are not under oral contraceptives should be aware that they are susceptible to increased gait instabilities in pre-menstrual phase after such activities.

### Funding
This work was supported by the Brazilian governmental agencies CAPES (Finance Code 001), CNPq, and FAPEG for supporting this study. Marcus Fraga Vieira is a fellow of CNPq (Process 304533/2020-3). The funders had no role in study design, data collection and analysis, decision to publish, or preparation of the manuscript.

### Grant Disclosures
The following grant information was disclosed by the authors:
Brazilian Governmental Agencies CAPES: 001.
CNPq: 304533/2020-3.
FAPEG.

### Competing Interests
Rafael Baptista, Ludmila Dos Anjos and Marcus Fraga Vieira are Academic Editors for PeerJ.

## Author Contributions

- Ludmila Dos Anjos conceived and designed the experiments, performed the experiments, analyzed the data, authored or reviewed drafts of the article, and approved the final draft.
- Fábio Rodrigues conceived and designed the experiments, performed the experiments, analyzed the data, prepared figures and/or tables, authored or reviewed drafts of the article, and approved the final draft.
- Sofia Scataglini analyzed the data, authored or reviewed drafts of the article, interpretation of results, and critical review of the text, and approved the final draft.
- Rafael Reimann Baptista analyzed the data, prepared figures and/or tables, authored or reviewed drafts of the article, interpretation of results, and critical review of the text, and approved the final draft.
- Paula Lobo da Costa analyzed the data, authored or reviewed drafts of the article, interpretation of results, and critical review of the text, and approved the final draft.
- Marcus Fraga Vieira conceived and designed the experiments, performed the experiments, analyzed the data, prepared figures and/or tables, authored or reviewed drafts of the article, and approved the final draft.

## Human Ethics

The following information was supplied relating to ethical approvals (i.e., approving body and any reference numbers):

All protocols were approved by the Local Ethics Committee for Human Research (protocol number 3.913.182), and the participants signed an informed consent form.

## Data Availability

The raw data are available in the Supplemental Files.

## Supplemental Information

Supplemental information for this article can be found online at http://dx.doi.org/10.7717/peerj.16223#supplemental-information.

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
