# Peer review of "Trunk variability and local dynamic stability during gait after generalized fatigue induced by incremental exercise test in young women in different phases of the menstrual cycle"

_PeerJ, doi:10.7717/peerj.16223_

## Round 0.1 · original submission · Major Revisions

Dear Authors
Your manuscript has been reviewed in depth by two specialized reviewers who raise several suggestions and recommendations on your manuscript.
The manuscript in its current state cannot be published, however I believe that the suggestions raised by the authors can be taken on board by you and we look forward to receiving a new manuscript with concrete answers for the reviewers.
There are aspects in the introduction that need to be clarified, and there are methodological aspects that both reviewers point out that need to be addressed in depth.

Reviewer 1 ·

Basic reporting

This is an interesting study, but there are many important areas for improvement.
- I do not understand the relationship between dehydration caused by general muscular exercise and impaired vestibular function. It would be interesting and necessary to add more references to support this assertion as only the one by Derave et al., 1998 was added and it is not very up to date.
- An exhaustive review of the literature is necessary as only 5 references out of 46 are from the last 5 years.
- Has the influence of mental fatigue during exercise been considered? There is a lot of literature on how mental fatigue can affect physical performance. Perhaps you should introduce a section where you discuss this and obviously include it in the limitations section.
- Too many acronyms are used and this can confuse the reader.
- In line 294 and line 303 you put "(table2)" in brackets when you are in the discussion. I recommend that in the discussion you do not refer to the tables and figures as they are in the results section.
- The limitations section is brief and the study has many limitations that have not been taken into accountand it would be interesting to expose them.
- The section on clinical implications can be introduced in a section at the end of the discussion.
- The tables are not well designed as they only show mean and variance data, but they do not show the data of the differences that are indicated in the text, so it is not possible to interpret the results.

Experimental design

- The structure of the article is not correct as important sections are missing, such as the study design, the variables and their description, they should be more specific in describing the test procedure and the sample size calculation should be added as part of the methods section.
- I do not understand the difference between the section on "data analysis" and "statistical analysis".
- Data analysis and statistical tests need to be further developed.
- I would put the hypotheses before describing the objectives.
- Despite the number of the ethics committee, the attached document is not an official document from an institution that guarantees that it has passed the ethics committee.

Validity of the findings

- The aim is not clear because by analysing only a sample of women both in the discussion and in the conclusion you indicate that variability increases and LDS decreases compared to men.

Reviewer 2 ·

Basic reporting

Thank you for giving me the opportunity to review this interesting manuscript.

The authors use a correct and understandable language for the reader. The manuscript, in general, is very extensive and there are parts in which the information can be schematized. We suggest that the introduction should not exceed two pages in length.
71-75. Authors should differentiate between the effects of high-intensity and high-volume exercise on dehydration.
During the introduction, it would be interesting to explain the effect (on fatigue) of hemoglobin/anemia combined with exercise in menstruating women.

Experimental design

The Figures and tables are relevant, high quality, well labelled & described; however, be sure to leave enough margins in the figure 3 for a correct reading of the figure.
Please add in table 1 the normality values of the Saphiro-WIlks test.
I recommend including the statistical data (not only descriptive) in Table 2 to properly read the changes produced.

112-115. Have patients with metabolic pathology, diabetes, oncology or anemia, also been excluded?

197. Note that the post-hoc was performed when differences were found in the ANOVA.

209-242. I believe that some of the information in the results could be summarized by including the statistics in Table 2.

Validity of the findings

148-154 It is very inaccurate to say that the threshold is above 60% of the maximum heart rate (it will depend mainly on ventilatory and lactate thresholds and these measurements were not performed in this study), especially when the HR is estimated (220-age). If a test has been done to the point of exhaustion, why not take the HR reached at the end of the test?

280-286. When comparing between genders or between women of different heights, for example, it is incorrect to say that a muscle is more efficient because it is smaller, since it also depends on ventricular size, lung size, blood volume, hematocrit, number and sizes of mitochondria... I suggest explain that also there are other factors that could affect muscle efficiency.

In view of the above, I believe that this research is novel, well written and with minor and correctable errors for its publication.

---

## Round 0.2 · Minor Revisions

Dear Authors
Thank you very much for addressing the previous reviewers' comments. In this new revision the reviewers acknowledge the efforts of the reviewers in addressing each suggestion, however one of the reviewers still raises minor changes that I believe need to be addressed.

We look forward to receiving a new version of your manuscript soon to consider a final acceptance.

Reviewer 1 ·

Basic reporting

Many thanks to the authors for forwarding the manuscript. It is a very interesting study and I congratulate you on the correction of most of the corrections.
Basic reporting:
- I appreciate the inclusion of new references supporting the relationship between dehydration caused by general muscular exercise and impaired vestibular function. It is very well integrated into the introduction and thus clearer for the reader.
- Regarding the timeliness of the references, I am grateful for the inclusion of some more current studies.
- The information included on the concept of mental fatigue is correct and sufficient both in the introduction and in the limitations.
- Thank you for the elimination of excessive acronyms. The IPAQ acronym could be kept as well as MC (Mentrual cycle) because the latter in particular is very useful.
- Thank you for removing "table 2" from the discussion and leaving it only in the results section.
- Thank you for expanding the limitations section of the study as it is of great interest to the reader.
- Regarding table 2, the pairwise comparison between the 3 phases of the menstrual cycle is not clear; FP-OP; FP-LP;LP-OP. It is necessary to clarify this in table 2 in order to know the difference between the phases.

Experimental design

Experimental design:
- -With regard to the study methods, perhaps I have not expressed myself well, but it is important that the first point under methods is entitled "study design" as it places the reader in a better position than if we directly set out the "participants" section, so I request the introduction of that section. As for the sample size, I agree with your argument and now I understand why this section has not been included.
- Thank you for clarifying the difference between "data analysis" and "statistical analysis".
- Thank you for including the correct ethics committee document and attaching the number of the ethics committee, as well as clarifying the main objective of the study in terms of gender comparison.

Validity of the findings

Thank you for the explanation regarding the objective. Still, despite the authors' explanation, from the reader's point of view I would rearrange the order of the objectives and the hypothesis. It makes more sense to first state a hypothesis and on the basis of that hypothesis describe the objective. As for the conclusions, they are well stated and answer the research objective.

Additional comments

I am grateful for the corrections. It is an interesting study and I think that by making the proposed changes this article could be published.

Reviewer 2 ·

Basic reporting

The authors have successfully made the changes suggested in this section.

Experimental design

The authors have successfully made the changes suggested in this section.

Validity of the findings

The authors have successfully made the changes suggested in this section.

---

## Round 0.3 · accepted · Accept

The original Academic Editor is not available so I am making the final decision ion this submission.

Thank you for addressing the remaining revisions to the manuscript.